# Establishment and Systematic Evaluation of Gastric Cancer Classification Model Based on Pyroptosis

**DOI:** 10.3390/diagnostics12112858

**Published:** 2022-11-18

**Authors:** Sultan F. Kadasah

**Affiliations:** Department of Biology, Faculty of Sciences, University of Bisha, P.O. Box 551, Bisha 61922, Saudi Arabia; sukadasah@ub.edu.sa

**Keywords:** gastric cancer, pyroptosis, inflammation, PCA, unsupervised cluster analysis

## Abstract

**Background:** Gastric cancer (GC) is considered the fifth most prevalent type of cancer and the third leading cause of cancer deaths worldwide. This in-depth investigation was performed to generate fresh concepts for the clinical classification, diagnosis, and prognostic evaluation of GC. **Methods:** The data were retrieved from the Gene Expression Omnibus (GEO) and The Cancer Genome Atlas (TCGA) databases. Unsupervised cluster analysis was used to divide up the GC patients using pyroptosis-related differentially expressed genes (DEGs), which were discovered to be significantly linked with GC prognosis. The therapeutic importance of pyroptosis in GC patients was discovered using PCA analysis of genes associated with pyroptosis. The models were then carefully scrutinized. **Results:** Three hub genes, ELANE, IL6, and TIRAP, exhibit significant predictive importance among the 15 pyroptosis-related genes. Unsupervised clustering analysis revealed that the DEGs were enriched in the pathway of cytokine–cytokine receptor interactions, and Clusters 1 and 2 had statistically distinct prognoses. PCA analysis revealed significant differences in the area under the curve, immunological checkpoints, immunogenic cell death, and prognostic value between the high- and low-risk groups. **Conclusions:** These two GC classification models, based on pyroptosis, have significant clinical value for patients with GC.

## 1. Introduction

The third greatest cause of cancer-related mortality and one of the top five most widespread malignancies is gastric cancer [1]. Globally, there were reportedly 1,089,103 new cases of GC in 2020 (5.6%) and 768,793 fatalities (7.7%) [2,3]. The frequency of GC occurrences has grown during the past 20 years. As a result, it has become critical to diagnose GC patients sooner, as recommended in Saudi Arabia [4]. Even after open standard gastrectomy or laparoscopic gastrectomy, the 5-year relapse-free survival percentage for patients with pathological stages III was only 45–55 percent, but the survival rate for those with stages I was 78–84 percent [5]. However, at the time of diagnosis, more than 60% of patients were in advanced stages [6]. The participation rate for gastroscopy is often less than 60%, despite the fact that it significantly increases the rate of early GC identification [7]. As a result, adopting more simple, safe, and efficient biological indicators to diagnose GC and estimate the prognosis of individuals with GC is critical [8].

Programmed cell death actively contributes to the preservation of homeostasis [9]. Pyoptosis, unlike apoptosis, is a pro-inflammatory type of programmed cell death. A plethora of cytokines are released, and a cascade of responses are triggered from the neighboring cells upon cell death [10]. Chemotherapy drugs (i.e., topotecan, etoposide, and cisplatin) can activate cysteinyl aspartate-specific proteinase 3 (caspase-3) and trigger pyroptosis by cleaving Gasdermin E (GSDME) [11]. Furthermore, Wang et al. demonstrated that 4T1 stage breast cancer might be completely cured when pyroptosis occurs in fewer than 15% of tumor cells [12]. Besides, pyroptosis was illustrated to regulate the tumor immune microenvironment via the BRAF and MEK inhibitors [13]. Therefore, the role of pyroptosis has been increasingly important in cancer development and treatment with advanced research.

Despite being important and promising, the exact role of pyroptosis has only been partially investigated in GC. In order to advance clinical categorization, diagnosis, and prognostic evaluation of GC, comprehensive research has been conducted.

## 2. Materials and Methods

### 2.1. Data Acquisition and Pre-Processing

The UCSC Xena database (https://xena.ucsc.edu/, accessed on 1 June 2020), which supports a variety of omics and clinical data, including gene-, transcript-, exon-, miRNA-, LncRNA-, protein-expressions, DNA methylation, and ATAC-se, was used to download the files for stomach adenocarcinoma (STAD) in Genomic Data Commons (GDC) and The Cancer Genome Atlas (for 32 normal samples and 350 GC samples, full transcriptome and survival data were collected in these files, and these data were used to find differentially expressed genes (DEGs) (HTSeq-counts file)) and conduct the study that followed (HTSeq-FPKM file). Gene annotations were then made using the genecode V22.annotation file.

### 2.2. Identification of Differentially Expressed Pyroptosis-Related Genes

A total of 33 genes were associated with pyroptosis, according to a thorough literature review [14,15,16,17,18]. Following the retrieval of these genes’ expression matrices from transcriptome data, normal and GC tissues were compared for pyroptosis-related DEGs using the ‘DESeq2’ tool. A threshold of adjusted *p*-value 0.01 was determined for significant pyroptosis-related DEGs. Both a heatmap and a volcano map were used to display the DEGs related to pyroptosis.

### 2.3. Unsupervised Cluster Analysis

Unsupervised clustering analysis is a popular method for tumor classification. In this work, we employed univariate cox regression analysis with a *p*-value of 0.10 to look for the genes associated with pyroptosis that were also associated with prognosis. Unsupervised clustering analysis was then utilized to classify the GC samples, based on the pyroptosis-related DEGs using the “ConsensusClusterPlus” tool. The distance was set to be “euclidean”, and the values for maxK, pTtem, REPS, and random seed were set to 7, 0.8, 1000, and 123456, respectively. Additionally, the resampling technique was used to extract data. By evaluating the rationality under multiple values of K between 2 and the maxK, the optimal clustering K value was identified (i.e., 7).

### 2.4. Immune Infiltration Analysis

A more recent variation of the GSEA is the single sample gene set enrichment analysis (ssGSEA). By using the ssGSEA to compare the gene expression data from the sample set with those from the provided gene set, the enrichment score was calculated. The higher the enrichment score, the greater the degree of enrichment. This study assessed the marker genes of 28 distinct types of immune cells, determined the enrichment scores of immune cells in each sample, and then conducted further analysis.

### 2.5. Pathway Enrichment Analyses

The hypergeometric distribution relationship between DEGs and certain GO branches, which identified genes and their underlying biological processes, was evaluated using Gene Ontology (GO). Each GO with DEGs were given a *p*-value, and the GO was enriched for the genes with the highest statistical significance. Important GO keywords and pathways were found using Fisher’s exact test [19], and the corrected *p*-value was obtained using the Benjamini and Hochberg false discovery rate method. The biological process, cellular component, and molecular function included in GO analysis had a suggestive impact on the results of experiments. The GO analysis of DEGs can reveal the gene function.

### 2.6. Prognostic Model

In order to minimize the dimensionality, principal component analysis (PCA) was carried out on patients with GC, based on the 33 pyroptosis-related genes. Orthogonal rotation and its interpretation were used to find the right dimensionality value, and the PCA establishment factor score was computed. Then, GC samples were divided into high- and low-risk groups based on the model’s median riskScore after a linear model based on the numerous principal component components, and scores of these dimensions were generated.

### 2.7. Model Evaluation

The prognosis disparity was investigated between high-risk and low-risk groups. Then, the mortality rates between the high- and low-risk groups were compared using the PCA establishment factor scores. The effectiveness and prognostic prediction precision of the model were assessed using the ‘timeROC’ software.

### 2.8. Quantitative Real-Time Polymerase Chain Reaction (qRT-PCR)

Total RNA was extracted from MGC-803 cell line, and complementary DNA (cDNA) was synthesized using total RNA with the PrimeScript™ RT reagent kit with gDNA Eraser (TaKaRa, Kusatsu, Japan), according to the manufacturer’s instructions. qRT-PCR was performed with AceQ Universal SYBR qPCR Master Mix (Vazyme, Nanjing, China) on an QuantStudio 7 PCR system (Thermo Fisher, Waltham, CA, USA). Primers used in this study were listed as follows: TIRAP (Forward): TCCACCAAAGAGAAAGCAGCC;TIRAP (Reverse): CTTCCTATGTAAGGCCGTAGTG.GAPDH (Forward): GGAGCGAGATCCCTCCAAAAT.GAPDH (Reverse): GGCTGTTGTCATACTTCTCATGG. Relative quantification was determined using the 2^−ΔΔCt^ method.

### 2.9. Statistical Analysis

The rank sum test was applied, in order to compare samples between two or more groups. Using R, the statistical analysis was completed (4.0.5 version). The statistical significance threshold was set at *p* < 0.05.

## 3. Result

### 3.1. Screening Pyroptosis-Related DEGs in GC

When DEGs associated with pyroptosis were sought using the ‘DESeq2’ program, 15 genes were discovered to have substantially different expression levels in GC and para-cancer tissues (*p* < 0.001). (Figure 1A). The top five pyroptosis-related DEGs were GSDMC, PRKACA, ELANE, CASP9, and NLRP6, whereas the top five down-regulated DEGs were CASP8, NLRC, AIM2, GSDMB, and NLRP2 (Figure 1B). The relationship between these genes was further examined using gene interaction analysis with absolute correlation values >0.2. These genes were shown to interact positively with numerous other genes, with the most significant interactions being between the genes NLRC4, AIM2, NLRP1, IL18, CASP8, and GSDMB (Figure 1C).

### 3.2. Pyroptosis-Related Subtypes in GC

In the univariate cox regression analysis of all 15 pyroptosis-related genes, only three hub genes—ELANE, IL6, and TIRAP—showed a *p*-value of less than 0.1. In the unsupervised clustering analysis, the ideal value of clustering K, based on the three hub genes, was 2, and there was a significant difference between the two sets of GC data (Figure 2A). The prognostic differences between the two pyroptosis-related subtypes of GC were subsequently examined using the Kaplan–Meier (KM) plotter, and patients in cluster 2 had a better prognosis than patients in cluster 1 (*p* < 0.05) (Figure 2B).

Additionally, the disparity in immune infiltration between the two pyroptosis-related subtypes of GC was examined, and the enrichment scores of 28 different types of immune cells in each sample were calculated using ssGSEA analysis. The enrichment scores in cluster 1 showed a rising trend, when compared to cluster 2. For all 18 types of immune cells, there were substantial changes in the enrichment scores between clusters 1 and 2, including the activated CD4 T cells, macrophages, memory B cells, and others (Figure 2C,D).

Additionally, GO and KEGG analyses of the DEGs of two pyroptosis-related subtypes of GC were performed. According to GO analysis, DEGs may be involved in the formation of the extracellular matrix, the organization of extracellular structures, the presence of collagen, the action of receptor ligands, and the activity of signaling receptor activators. The PI3K-Akt signaling pathway, neuroactive ligand-receptor interaction, and cytokine–cytokine receptor interaction were enriched by KEGG pathway analysis (Figure 2E,F).

### 3.3. Constructing and Evaluating a Pyroptosis-Related Prognostic Model of GC

Based on the 33 genes connected to pyroptosis, the PCA approach was used. The eigenvalues fall and level out as the component increases past four (Figure 3A), and the proper dimensionality value was calculated. The four main component variables (i.e., RC1-RC4) and their scores were then used in a linear regression analysis to compute the riskScore. In both the TCGA and GSE62254 cohorts, patients who passed away had riskScores that were significantly higher than those who survived (*p* < 0.05). (Figure 3B,C). Additionally, as seen in Figure 3D,E, individuals with GC were more likely to die when their riskScore was greater.

The prognostic model was also evaluated. To distinguish between the high-risk and low-risk categories, the median riskScore was employed (Figure 4A,B). Using T-SNE, a non-linear dimensionality reduction technique [20], the ability of the pyroptosis-related prognostic model to distinguish between GC patients with high- and low-risk was evaluated. Considerable discrimination was found (Figure 4C,D). Additionally, in both cohorts, the high-risk group’s PCA establishment factor scores for RC1, RC2, and RC3 increased relative to the low-risk group’s scores, with the exception of RC4, which was higher in the low-risk group (Figure 4E,F). Additionally, patients in both cohorts who had lower riskScores fared better, in terms of survival, than those who had higher riskScores (*p* < 0.05). (Figure 4G,H). Then, using the “timeROC” tool and the receiver operating characteristic curve, the precisions of the clinical indicators and riskScore for prognostic prediction of GC patients were assessed (ROC). The values of the area under the curve (AUC) were obtained, and they have been stable over the last five years. The AUC in the TCGA cohort was around 0.65, while in the GSE62254 cohort, it was roughly 0.7. The accuracy was significantly increased by a thorough analysis of clinical characteristics and riskScores (AUC was over 0.7) (Figure 4I,J).

### 3.4. Immune Infiltration Analysis

Cancer growth and treatment are significantly influenced by immune checkpoints (ICPs) and immunogenic cell death (ICD) genes [21,22]. Based on differences between the pyroptosis-related high- and low-risk groups, most ICPs-related genes, including as CD44, CTLA4, and TIGIT, were up-expressed in the high-risk group in both the TCGA and GES62254 cohorts (Figure 5A,B). Surprisingly, in both cohorts, the majority of ICD-related genes were up-regulated in the high-risk group. In the TCGA cohort, the high-risk group had higher levels of expression of HIGF, EIF2AK2, P2RX7, PANX1, EIF2AK4, TLR4, IFNAR1, IFNAR2, and FPR1, while EIF2AK1, MET, EIF2A, CALR, and HMGB1 were lower (*p* < 0.05) (Figure 5C). In the GES62254 cohort, the high-risk group significantly expressed HGF, EIF2AK2, P2RX7, PANX1, EIF2AK4, TLR4, IFNAR1, IFNAR2, and FPR1, whereas the MET, EIF2A, CALR, and HMGB1 levels were similarly decreased. (Figure 5D).

Using the “MCPcounter” program, the scores of the 10 different types of immune cells in each sample were determined. The riskScores of the model showed a strong correlation with these immune cells in the two cohorts, particularly in fibroblast and endothelial cells (Figure 5E,F).

### 3.5. Conducting Nomograms

Figure 6A,B illustrates the relationship between RiskScore and age and the prognosis of patients with GC. This model demonstrated a prognostic relevance. At 1, 3, and 5 years, the TCGA cohort’s patients with TCGA-VQ-AA6A had mortality rates of 20, 5, and 63.9 percent, respectively, according to this model. In the GSE62254 cohort, the mortality rates for GSM1523986 patients were 26.1 percent, 65.3 percent, and 75.6 percent at 1, 3, and 5 years, respectively. Additionally, the nomogram-predicted overall survival at 1, 3, and 5 years in the two cohorts had notable prediction values (Figure 6C,D). AUC values were also 0.69, 0.71, and 0.73 for the TCGA cohort at 1, 3, and 5 years and 0.80, 0.77, and 0.74 for the GSE62254 cohort, respectively. This shows that, among GC patients, nomograms had a strong prognostic value (Figure 6E,F).

### 3.6. PCR Was Performed to Verify the Expression of TIRAP

We collected gastric cancer cell line MGC-803 and performed PCR experiments to verify the expression of the key gene TIRAP (Figure 7). The results showed that TIRAP was significantly up-regulated in MGC-803 cell line.

## 4. Discussion

The former categorization of cell death was apoptosis and unintentional necrosis [23]. However, since the fast cell death based on caspase-1 activity was observed in macrophages in 2001 [24], pyroptosis has been a unique type of programmed cell death. Pyroptosis and inflammation have a close relationship, and the cysteine protease procaspase1 and NLRP3 can be combined to create inflammasomes [25]. Additionally, proinflammatory cytokines, including interleukin (IL) 1 and IL18, are produced by cells going through pyroptosis, which is brought on by gasdermin D (GSDMD) [26]. Additionally, pro-inflammatory cytokines can stimulate immune cells, which intensifies the inflammatory response [27]. As a result, pyroptosis, which can be defined as a type of programmed cell death, has a tight relationship with inflammation and immunological activation.

Pyroptosis and GC have also been connected. As a crucial element of pyroptosis, GSDMD expression decreased in GC cells, in comparison to neighboring non-cancerous cells [28]. A total of 33 pyroptosis-related genes were assembled from comparable studies in our work, and 15 DEGs between GC and nearby tissues were examined for each of the 33 pyroptosis-related genes. Then, using a cox regression model, the association between these 15 genes and the prognosis was evaluated. Additionally, the hub genes ELANE, IL6, and TIRAP were discovered, which greatly improved the prognosis. These three hub genes associated with pyroptosis were crucial for both pyroptosis and GC. Inflammatory reactions have a role in the GC’s development, including the start, growth, and metastasis of the tumor [29]. IL-6 is highly up-regulated in GC, and associated with the unfavorable prognosis [30] and poor responsiveness to chemotherapy [31].

The GC patients were then divided into two groups, high-risk and low-risk groups, using an unsupervised cluster analysis carried out on the hub genes. This differentiates patients with GC using a novel classification system based on genes associated with pyroptosis. The classification system may distinguish between individuals with a poor prognosis and those with a bright outlook, according to further studies. The expression of cells associated with immunity and inflammation varied significantly across the two groups, as well. These results demonstrated that this classification strategy is beneficial in identifying GC patients with high and low risks. In order to search for potential routes, the DEGs between the two groups were also enhanced. Our results not only provide a reference for the prognostic stratification of GC, but also provide a method for assessing the immune status of patients.

In order to determine the clinical importance of pyroptosis in GC patients, PCA analysis of the genes associated with pyroptosis was also performed. After selecting the appropriate dimensionality value using orthogonal rotation and its interpretation, the PCA establishment factor score was produced. To minimize dimensionality, a linear regression model has been used to form the riskScore. After dividing the patients into high-risk and low-risk groups using the median riskScore, the clinical value was evaluated. This quantitative classification method was based on PCA analysis and genes relevant to pyroptosis. These data imply that pyroptosis-related genes have a significant impact on the GC, and additional research into the process is warranted. We also provide a cutting-edge strategy for GC diagnosis and treatment.

ICPs are only one method used by cancer cells to elude the immune system’s attack. This strategy allows cancer cells to hide throughout the human body [32,33]. Nivolumab and pembrolizumab are examples of immune checkpoint inhibitors (ICIs) that have lately gained attention as prospective cancer treatment options [34]. There has been some research on the connection between pyroptosis and cancer, but nothing is known about the relationship between ICPs and pyroptosis. Our study used pyroptosis-related genes to categorize participants into high- and low-risk groups, and the expression of ICPs varied considerably between these groups. This demonstrated the close connection between ICPs and pyroptosis. Additionally, cancer cell death can be immunogenic or nonimmunogenic [35], and our findings propose that pyroptosis-related genes might control ICD.

In conclusion, these two GC classification models based on pyroptosis have significant clinical value for patients with GC.

## Figures and Tables

**Figure 1 diagnostics-12-02858-f001:**
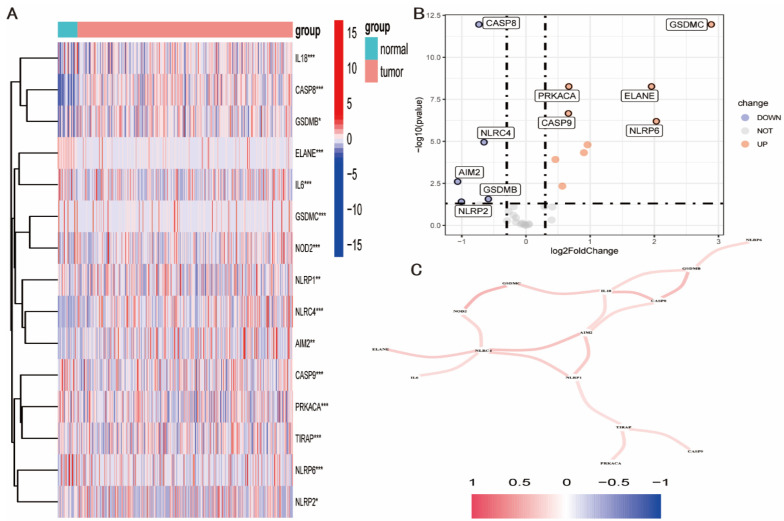
Screening pyroptosis-related DEGs in GC. (**A**) Heat map of pyroptosis-related differentially expressed genes between GC and para-cancer tissues. * 0.05; ** 0.01; *** 0.001. (**B**) Volcano map of pyroptosis-related differentially expressed genes between GC and para-cancer tissues. (**C**) Gene interaction analysis was conducted on pyroptosis-related differentially expressed genes with absolute correlation value > 0.2.

**Figure 2 diagnostics-12-02858-f002:**
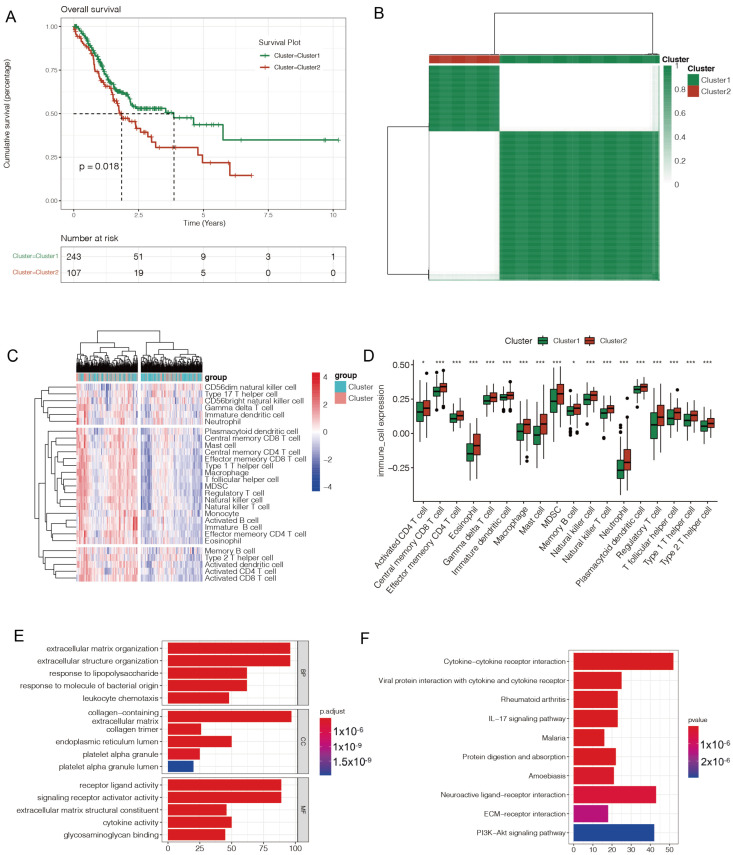
Pyroptosis-related subtypes in GC. (**A**) The unsupervised clustering analysis was performed in the patients with GC based on 3 hub genes, including ELANE, IL6, and TIRAP. (**B**) The Kaplan–Meier plotter was employed to investigate the prognostic difference of patients with GC between cluster 1 and cluster 2. (**C**) The enrichment scores showed an upward trend in cluster 1, compared with cluster 2. (**D**) In all 18 types of immune cells with significant differences, the enrichment scores were higher in cluster 1 than cluster 2. * 0.05; *** 0.001. (**E**,**F**) The GO and KEGG analysis were conducted in the DEGs of two pyroptosis-related subtypes of GC.

**Figure 3 diagnostics-12-02858-f003:**
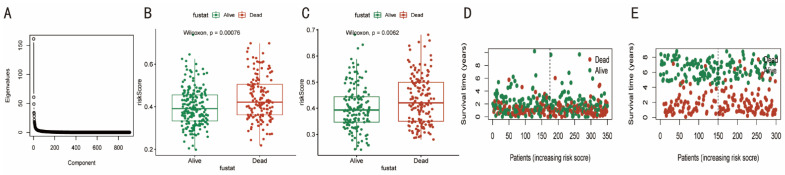
Constructing the pyroptosis-related prognostic model of GC. The PCA method was performed based on all 33 pyroptosis-related genes. (**A**) When the component is beyond four, the eigenvalues decline, leveling off. (**B**,**C**) In both TCGA and GSE62254 cohorts, the riskScores were significantly higher in patients who died than in those who survived. (**D**,**E**) In both TCGA and GSE62254 cohorts, the mortality in patients with GC tended to be increased with the higher riskScore.

**Figure 4 diagnostics-12-02858-f004:**
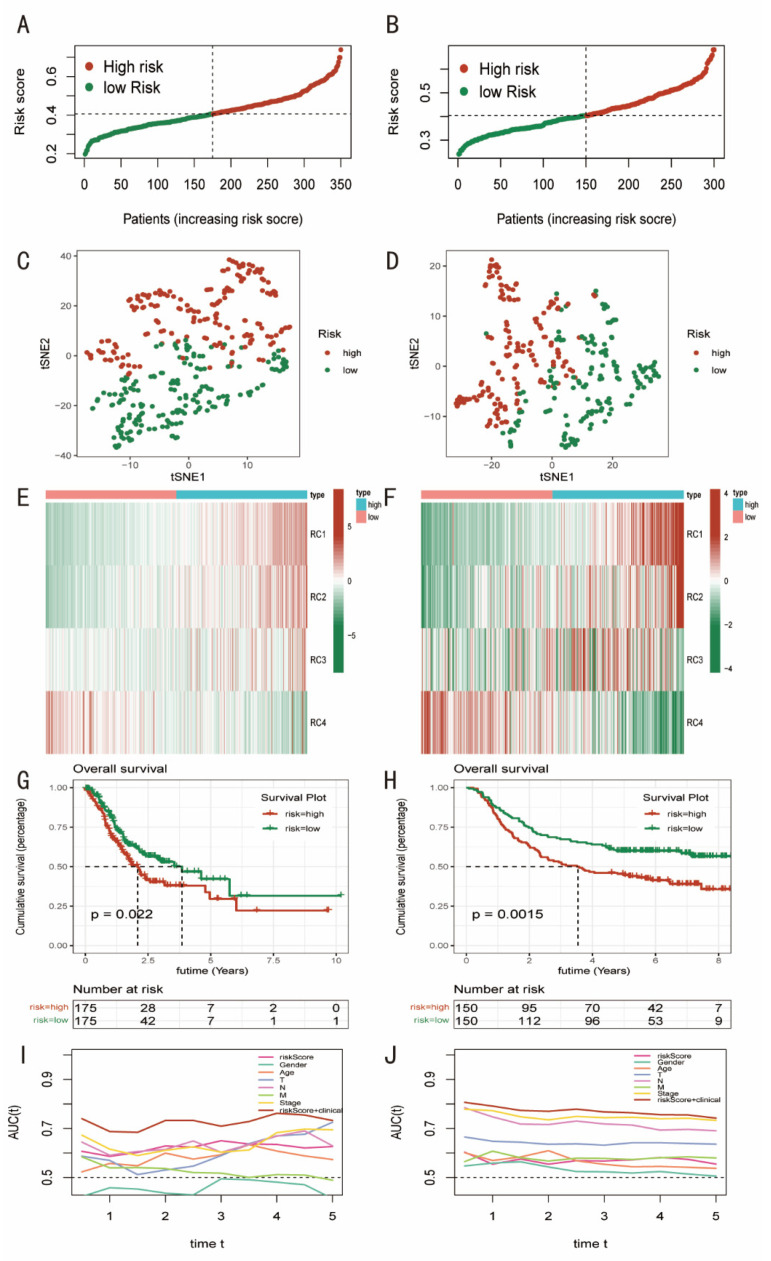
Evaluating the pyroptosis-related prognostic model of GC. (**A**,**B**) The patients with GC were divided into high- and low-risk groups, based on the median riskScore. (**C**,**D**) T-SNE was applied to assess the ability of the pyroptosis-related prognostic model to distinguish between GC patients with high- and low-risk. (**E**,**F**) The PCA establishment factor scores of RC1, RC2, and RC3 increased in the high-risk group than in the low-risk group, while the score of RC4 was higher in the low-risk group. (**G**,**H**) Patients in the low-risk group had improved survival, compared with those in the high-risk group. (**I**,**J**) The area under curve (AUC) values was obtained, which is stable for five years.

**Figure 5 diagnostics-12-02858-f005:**
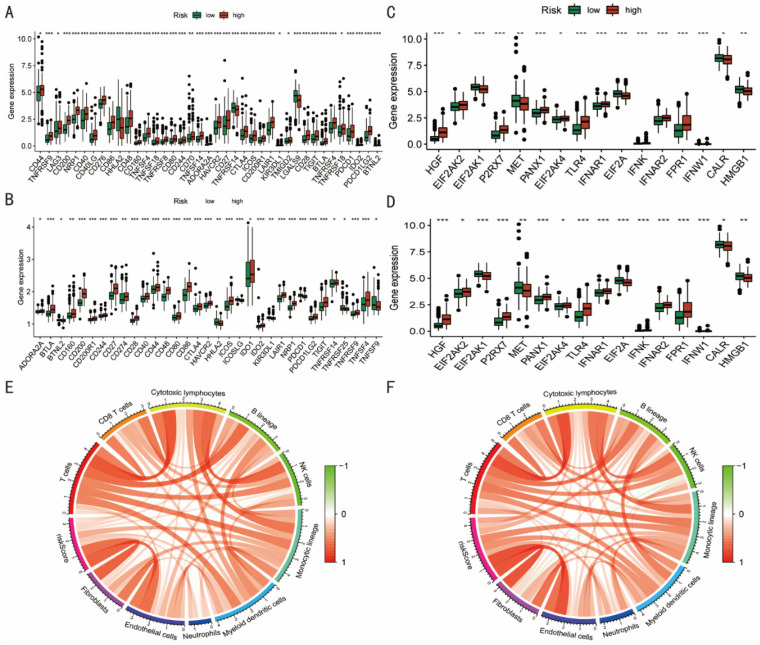
Immune infiltration analysis. (**A**,**B**) Most immune checkpoints-related genes were up-expressed in the pyroptosis-related high-risk group than low-risk group. (**C**,**D**) Immunogenic cell death-related genes mainly showed up-expression in the high-risk group. * 0.05; ** 0.01; *** 0.001. (**E**,**F**) The riskScores of the model were positively correlated with these immune cells.

**Figure 6 diagnostics-12-02858-f006:**
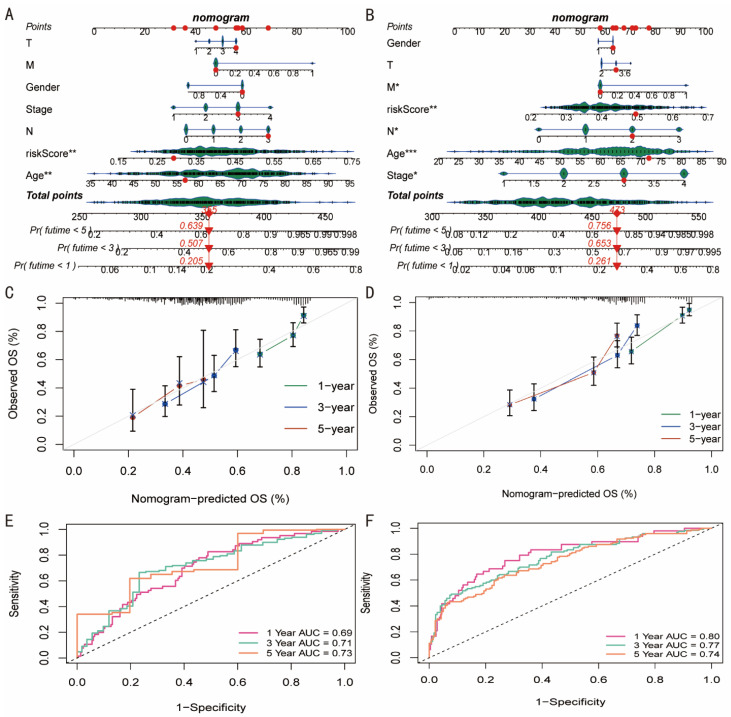
Conducting nomograms. (**A**,**B**) RiskScore and age were associated with the prognosis of patients with GC, and nomograms were conducted. * 0.05; ** 0.01; *** 0.001. (**C**,**D**) Nomogram-predicted OS at 1, 3, and 5 years were conducted. (**E**,**F**) The AUC at 1, 3, and 5 years were calculated.

**Figure 7 diagnostics-12-02858-f007:**
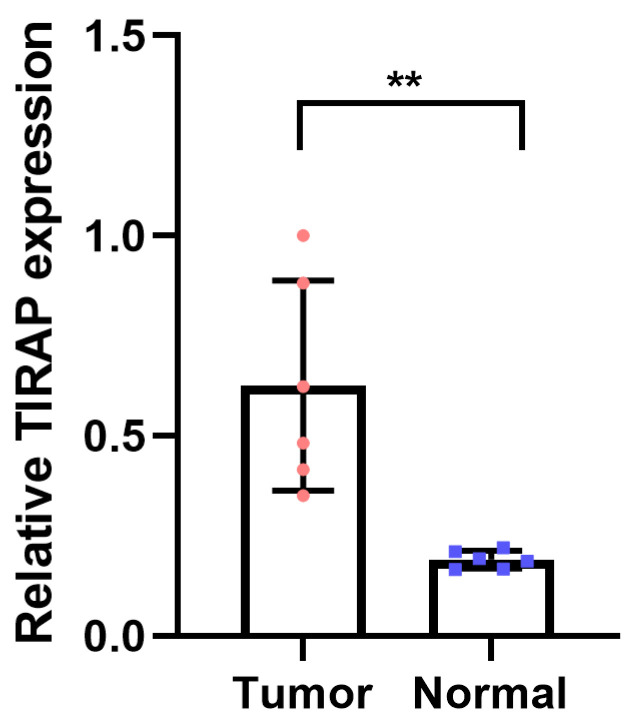
PCR was performed to verify the expression of TIRAP (** *p* < 0.01).

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
