# Peer review of "Establishment and Systematic Evaluation of Gastric Cancer Classification Model Based on Pyroptosis"

_diagnostics, 2022, doi:10.3390/diagnostics12112858_

Round 1

Reviewer 1 Report

1. As the author mentioned “32 normal samples and 350 GC samples, full transcriptome and survival data were collected” in the database. I wonder whether the patients have different diagnosis strategies, which probably influence both gene expression pattern and survival rate. How did the author consider about the effect of clinical treatment to the classification models ?

2. I suggest the author verify the up and down regulated key gene expression in gastric cancer cells compared with normal ones. The quantitative analysis of gene expression would be rapid and convincing.

3. More and more targeted drugs for gastric cancer are emerging now. Is any of them related to pyroptosis pathways ?

4. I wonder whether the two GC classification models have preference for certain areas and genome background.  

Author Response

Uploaded 

Reviewer 2 Report

The authors want to reveal the correlation between the classification of gastric cancer based on cell pyroptosis and the clinical diagnosis and prognosis of patient. However, the current study lacks novelty, and contains several major problems.

 Limitations:

1. The number of cases included in the study was small. The difference between 32 normal samples and 350 GC samples was large.

2. The authors need to use clinical samples to verify the conclusions from database analysis.

3. The authors need to further explain the clinical significance of this classification model, such as how to guide clinical treatment and how to use the model to predict the prognosis of patients.

Author Response

Uploaded

Reviewer 3 Report

In this manuscript, the authors analyze the association of pyroptosis with gastric cancer classification. They found that three pyroptosis genes: ELANE, IL6, and TIRAP, exhibit significant predictive association. Clustering analysis revealed that the DEGs were enriched in the pathway of cytokine-cytokine receptor interaction and that Clusters 1 and 2 had statistically distinct prognoses. PCA analysis revealed significant differences in the area under the curve, immunological checkpoints, immunogenic cell death, and prognostic value between the high and low-risk groups. At last, they conclude that the classification of GC based on pyroptosis signature has significant clinical value. However, the small sample size and small pyroptosis signature gene cluster may not fully support the association of pyroptosis with GC classification.

Author Response

Uploaded

Round 2

Reviewer 2 Report

The manuscript has been improved.

Reviewer 3 Report

I think the authors have answered all the comments nicely. 
I recommend the manuscript be published in Diagnostics.